# Predictors of ambulance transport to first health facility among injured patients in southern Sri Lanka

**Lindy M. Reynolds**[1]*, **Vijitha De Silva**[1,2], **Shayna Clancy**[3], **Anjni Joiner**[1,4], **Catherine A. Staton**[1,4], **Truls Østbye**[1,3]

1 Duke Global Health Institute, Duke University, Durham, NC, United States of America, 2 Faculty of Medicine, University of Ruhuna, Galle, Sri Lanka, 3 Department of Family Medicine and Community Health, Duke University School of Medicine, Durham, NC, United States of America, 4 Department of Surgery, Duke University School of Medicine, Durham, NC, United States of America

* lindyreynolds7@gmail.com

**Data Availability Statement:** All relevant data are within the manuscript and its Supporting Information files.

## Abstract

### Background

Injuries account for about 13% of all registered deaths in Sri Lanka and are the leading cause of admission to public hospitals. Prehospital trauma care is new to Sri Lanka, and in 2016, a free ambulance service was launched in the Western and Southern provinces.

### Objective

The aim of this study was to identify the proportion of admitted injury patients at a tertiary hospital who used an ambulance to get to the first health facility and examine patient demographics, injury event, and injury type as predictors of ambulance transport.

### Methods

A cross-sectional survey was administered to 405 patients who were admitted to the emergency trauma center at Teaching Hospital Karapitiya (THK) in Galle, Sri Lanka. Descriptive statistics were tabulated to summarize prehospital transportation variables. Logistic regression models were created to examine predictors of ambulance transport, and ArcGIS Pro was used to calculate the distance between injury location and first facility and THK.

### Results

The proportion of patients with injuries who used an ambulance to get to the first health facility was 20.5%. Factors that were significantly associated with ambulance use were older age, injury mechanism, alcohol use prior to injury, location type, open wound, abrasion, and chest/abdomen injury. Distance from injury location to THK or nearest health facility were not significantly associated with ambulance transport to the first health facility.

### Conclusion

Among lower acuity injury patients in southern Sri Lanka, 20.5% traveled in an ambulance to the first health facility, while over half used a tuk tuk. Older age and injuries at home were

**Funding:** The author(s) received no specific funding for this work.

**Competing interests:** The authors have declared that no competing interests exist.

associated with lower odds of ambulance transport. Future studies on predictors of ambulance transport should include patients with more severe injuries, gather detailed data on care provided while in transport and examine the association between prehospital care and clinical outcomes.

## Introduction

Globally, approximately 5 million people die each year from injuries, which accounts for 9% of all deaths with most of these occurring in low and middle income countries [1]. In Sri Lanka, self-harm, road traffic injury (RTI), and interpersonal violence are the second, eighth, and ninth leading cause of years of life lost (YLLs) respectively [2]. In addition to fatal injuries, non-fatal injuries contribute to the overall disease burden and are the leading cause of hospital admissions in Sri Lanka with approximately 62,377 non-fatal injuries requiring in-patient care each year in the Galle district alone [3,4]. While the incidence and burden of different types of injuries in Sri Lanka is well documented through previous studies [2–10], far less is known about what occurs between injury event and arrival at the hospital.

Prehospital emergency care is relatively new in Sri Lanka, and there is wide variation in how patients arrive to the hospital. Consequently, there is considerable variation in where and the type of care patients receive before arriving at the hospital. Patients use commercial, private, and non-motor vehicles to get to the hospital [8,11–13]. Efforts to develop a prehospital system in Sri Lanka began in 2003 through a collaboration between government agencies and private companies; however, this Emergency Medical Services (EMS) pilot project was halted after the tsunami struck in December of 2004 [14]. In 2005 and continuing until 2016, many attempts were made by commercial ambulance companies, hospital-based services, volunteer services, and the fire department service to establish a prehospital system in Sri Lanka, but these efforts were not coordinated or centralized [13,14]. During this period, there were no consistent training standards for emergency medical technicians (EMTs) between ambulance services, a lack of public awareness about proper use of ambulance services, and no centralized communication [13,14]. Through a partnership with Medical Teams International and the Ministry of Health Sri Lanka and Trauma Secretariat, the northernmost district, Jaffna, implemented an EMS system in February of 2009 [12,14]. In addition to the partnership, the regional director of health services contributed resources that enabled the development of an EMS system in Jaffna [14]. The system in Jaffna responded to 2,124 incidents in the first 11 months of operation [12]. In 2016, the government of India awarded a grant to Sri Lanka to help develop an ambulance service with a unified emergency number (#1990) that would serve the people in the Western and Southern provinces for free. The service would employ 250 advanced life support (ALS) trained EMTs who undergo intensive training for three months in India in addition to 250 drivers who are trained in basic first aid to handle minor injuries or other traumas [13–15]. Traffic laws regarding siren and light usage as well as an ambulance's right of way were passed soon after the ambulance service was established. Ambulances are widely considered to be one of the fastest modes of transport to the hospital; however, there is a delay between when the ambulance is called and when it arrives at the injury location where a tuk-tuk might be more readily available to transport the person to the hospital. While initial funding for the ambulance service was provided by the Indian government, the ambulance service is now overseen by the Sri Lankan Society of Critical Care and Emergency Medicine and the next phase of implementation was scheduled for 2018 which involves the addition of 200 ambulances and expansion to other provinces [14].

As the ambulance service has only recently been established, and may still not be equally available in all areas, the demographics and injury characteristics of those who are using ambulances to travel to the first health facility as well as the locations of where the ambulance is being dispatched to are not well understood. Thus, the current study sought to identify the proportion of patients who used an ambulance to arrive to the first health facility and to assess demographic, injury characteristics as predictors of ambulance transport to the first health facility.

## Methods

### Study design, participants, setting

This was a hospital-based, cross-sectional study which consisted of a questionnaire administered to patients admitted to the Victoria Emergency Trauma Center (ETC) of Teaching Hospital Karapitiya (THK) located in Galle, Sri Lanka plus extraction of information from medical records. Patients were considered eligible for participation in the study if they were 18 years or older, presented with an externally-caused, acute injury that was severe enough to require overnight inpatient care, and if they could respond to the survey questions themselves. THK is the largest tertiary health care facility in the Southern Province with 1560 beds on 54 wards and several specialty units [16], and serves the population of the Southern province and the surrounding areas [16]. In Sri Lanka, healthcare is predominantly provided by government with the Ministry of Health operating 631 health facilities throughout the country [16]. Within the public hospital network there are three tiers of facilities which include primary, secondary and tertiary care institutions [17,18].

### Procedures and data collection

The strategy for sampling and recruiting subjects from the target population was similar to methods used in previous studies on prehospital surveillance in resource-limited settings [19–22]. Data collection for this study was hospital-based, and study staff coordinated with hospital staff to minimize disruptions to the patients' care [19–22]. Convenience sampling was used to approach any patient who met inclusion criteria about participation in this study. Each day of data collection, the research assistants would obtain a list of eligible patients from the emergency trauma center and short stay ward registries. Patients were approached individually and if they agreed, the research assistant would administer the paper questionnaire. If the hospital staff indicated that the patient's condition was so severe that he or she was not capable of providing consent, the research staff did not approach them. Data from the medical records were extracted daily for relevant information regarding the consented patients' injury and medical treatment.

Our primary method of data collection was through a questionnaire administered by native research assistants fluent in English (S2 File) and Sinhalese. The questionnaire was developed through combining the World Health Organization's Injury Surveillance form, Surgeons OverSeas Assessment of Surgical Need Version 3.0, and two other questionnaires from other studies that had similar objectives but were conducted in different settings [19,23–25]. Survey questions were adapted to the local context after input from local physicians. Following completion of the questionnaire, it was translated to Sinhalese. The questionnaire had five components: patient demographics, injury event, traffic injury, prehospital transportation and care, and a data collection form. The first four sections were completed via a brief interview of the patient, while the data collection form was completed by extracting relevant information from the medical records.

### Ethics approval

This study received approval from both the Ethics Review Committee at the University of Ruhuna, Faculty of Medicine in Sri Lanka and from the Duke University Institutional Review Board.

### Statistical analysis

Data was collected and then entered into a password protected electronic database using RED-Cap [26]. Following initial data entry, questionnaires were checked a second time for any errors that occurred during the original entry. Data analysis was carried out using Microsoft Excel (Microsoft Corporation, Redmond, WA, USA), Stata 15.0 (College Station, TX, USA), and ArcGIS Pro (ESRI, Redlands, CA, USA). Proportions and frequencies were used to describe the study sample, the injury event, and any categorical prehospital transportation and care variables. Means and standard deviations were used to summarize continuous variables. Mode of transport used to travel to first health facility included tuk-tuk, car or van, ambulance, bus, motorbike, truck, tractor, pedestrian and other, but was coded as binary variable (ambulance versus other mode of transport) for this analysis.

To determine which factors were associated with whether a patient used an ambulance to get to the first health facility, a series of logistic regression models were created with patient demographics, injury event and injury type, and body part injured as predictor variables. For bivariate models significance was set at $p<0.1$. Two multivariable models were created to examine whether patient demographics, injury event and injury type characteristics were significantly associated with whether a patient used an ambulance to get to the first health facility. The first model included injury mechanism, alcohol, location type, injury type and body part injured as predictors, and the fully specified model included age and gender in addition to all the variables in the first model.

To better understand whether distance from site of injury to THK or nearest health facility was associated with use of ambulance transport to the first health facility, injury locations and health facilities were geocoded to obtain longitude and latitude coordinates. Only health facilities in the Western, Southern, Sabaragamuwa, and Uva provinces were geocoded, as these were the provinces relevant to our study. An XY event layer was created from the coordinates, and a geographic transformation was performed. The point feature class was reprojected in the Kandawala Coordinate System. The Generate Near Table tool was used to obtain distances from injury locations to THK and the nearest health facility. Distance to THK and distance to nearest health facility were then entered in to logistic regression models to determine if it was significantly associated with ambulance transport to first health facility. In all logistic regression models, use of an ambulance to get to the first health facility was the dependent (outcome) variable.

### Results

During data collection, 405 patients who were admitted to the emergency trauma center of THK enrolled in our study. Over half were male 236 (236, 58.3%), and the average age was 44.5 (SD = 17.2) years. More worked at an inside desk job or were a shop worker (142, 35.1%) than any other reported job type (Table 1). The most common injury mechanism reported was road traffic injury (168, 41.5%) followed by falls (116, 28.6%) (Table 2). Alcohol use within six hours of the injury event was reported by 18.3% of participants.

Stage 1 refers to the segment of the prehospital journey from injury site to first health facility which could have been THK or another hospital. More than half (233, 57.5%) of the patients in this study came directly to THK from the injury site. The top three modes of

**Table 1. Patient characteristics: Frequencies and proportions.**

| | Overall (n = 405) | |
|---|---|---|
| **Patient Characteristics** | n | % |
| **Gender** | | |
| Male | 236 | 58.3 |
| Female | 169 | 41.7 |
| **Age (years)** | | |
| 18 to 39 | 191 | 47.2 |
| 40 to 55 | 111 | 27.4 |
| 56 to 95 | 103 | 25.4 |
| **Job** | | |
| Housewife | 43 | 10.6 |
| Outside/Manual Labor | 93 | 23 |
| Inside desk Job/Shop Worker | 142 | 35.1 |
| Other | 49 | 12.1 |
| Unemployed | 78 | 19.3 |
| **Monthly Income in Rupees (LKR) (45000 LKR = 242 USD)** | | |
| < = 45,000 | 279 | 68.9 |
| >45,000 | 126 | 31.1 |
| **Household Vehicle** | | |
| Bicycle | 21 | 5.2 |
| Motorbike | 192 | 47.4 |
| Tuk-tuk | 84 | 20.7 |
| Car | 58 | 14.3 |

transportation used to get to the first health facility were tuk-tuk, car or van, and ambulance (Table 3). In the current study, 20.5% of the 405 patients used an ambulance to get from the injury site to the first health facility (Fig 1). The geographic coordinates for the injury locations of 5 patients could not be obtained; thus, logistic models where distance was included as a predictor are based on a sample size of 400. For stages two and three, ambulances were used almost exclusively for transfer of patients between hospitals, and for this reason the analyses focus on the first stage.

Age, falls, stab or cut, other blunt force, alcohol use within six hours of the injury event, having been injured at home, open wound, abrasion, and chest or abdomen injury were all significantly associated with using an ambulance to get to the first health facility. Neither distance from injury location to THK or distance to the closest health facility were significantly associated with ambulance transport to the first health facility (Fig 1). Patients in the 56 to 95 age group facility were about 58% less likely (OR = 0.42, 95% CI 0.22–0.81) to use an ambulance to get to the first health facility than those the 18 to 39 age group (Table 4). Falls, stab or cut, and other blunt force were all associated with lower odds of using an ambulance during stage 1 of the prehospital trip compared to those injured in a RTI, while the odds of using an ambulance for those who were injured by a stab or cut were the lowest (OR = 0.28, 95% CI 0.10–0.75). Patients injured at home had 74% lower odds (OR = 0.26, 95% CI 0.14–0.48) of taking an ambulance to the first health facility compared to those injured in the street (Table 4). Open wound and abrasions were associated with increased odds of using an ambulance, compared to not having those respective injury types (Table 4).

Two multivariable models were created to assess predictors of use of an ambulance to get to the first health facility, and the results of the fully specified model are reported in Table 4. In

**Table 2. Injury characteristics: Frequencies and proportions.**

| Injury Characteristics | | |
|---|---|---|
| **Injury Mechanism** | | |
| RTI | 168 | 41.5 |
| Fall | 116 | 28.6 |
| Stab or Cut | 45 | 11.1 |
| Other Blunt Force | 57 | 14.1 |
| Other | 19 | 4.7 |
| **Alcohol** | | |
| No | 331 | 81.7 |
| Yes | 74 | 18.3 |
| **Location Type** | | |
| Street | 180 | 44.4 |
| Home | 143 | 35.3 |
| Work | 48 | 11.9 |
| Market | 10 | 2.5 |
| Other | 24 | 5.9 |
| **Injury Type[1]** | | |
| Open Wound | 50 | 12.4 |
| Abrasion | 66 | 16.3 |
| Fracture | 228 | 56.3 |
| Laceration | 165 | 40.7 |
| **Body Part Injured[1]** | | |
| Head/Face | 122 | 30.1 |
| Neck | 11 | 2.7 |
| Upper Limb | 164 | 40.5 |
| Chest/Abdomen | 39 | 9.6 |
| Spine | 24 | 5.9 |
| Lower Limb | 194 | 47.9 |

[1]Categories are not mutually exclusive. A patient could have more than one injury type and more than one body part injured. % is calculated based total study sample (n = 405).

the first model, odds of using an ambulance to get to the first health facility were 94% higher among those who reported alcohol use within six hours of the injury event compared to those who did not (OR = 1.94, 95% CI 1.04–3.62; p = 0.027). Alcohol use did not significantly predict use of an ambulance in the fully specified model (Table 4). Compared to injuries that occurred in the street, injuries that occurred at home had significantly lower odds of ambulance use in both models.

## Discussion

This study examined characteristics of the prehospital transportation and care of 405 patients with injuries that were admitted to the emergency trauma center at THK. Patient demographics and injury event were assessed as predictors of ambulance transport to the first health facility. The results indicate that 20.5% of patients used an ambulance to get to the first health facility.

While patient characteristics and injury event characteristics found in this study sample align with previous studies, characteristics of prehospital transportation and care were different from previous studies. These differences may be attributed to the health care delivery

**Table 3. Prehospital transportation and care characteristics.**

| | Overall (n = 405) | |
|---|---|---|
| **Prehospital Transportation and Care** | **n** | **%** |
| **>1 hr Delay in Seeking Care** | | |
| No | 345 | 85.2 |
| Yes | 60 | 14.8 |
| **Stage 1 Mode** | | |
| Tuk-tuk | 213 | 52.8 |
| Car or Van | 87 | 21.5 |
| *Ambulance* | 83 | *20.5* |
| Motorbike | 15 | 3.7 |
| Bus | 3 | 0.7 |
| Truck | 1 | 0.3 |
| Pedestrian | 1 | 0.3 |
| Other | 2 | 0.4 |
| **Paid for Transport** | | |
| No | 308 | 76.1 |
| Yes (Mean = 556 LKR \| 3 USD) | 94 | 23.2 |
| Missing | 3 | 0.7 |
| **Prior Medical Care** | | |
| No | 171 | 42.2 |
| Yes | 234 | 57.8 |
| **# of Stages of PH Trip** | | |
| 1 | 233 | 57.5 |
| 2 | 156 | 38.5 |
| 3 | 16 | 4 |
| **Transport Time by Stage (min)** | **Mean** | **SD** |
| Stage 1 | 27 | 18.3 |
| Stage 2 | 55 | 41.1 |
| Stage 3 | 72 | 38.3 |

infrastructure or the more developed EMS system [13,27,28]. Ambulance transport to first health facility was markedly higher in this study (20.5%) than in similar studies conducted in different resource-limited settings [29–31]. It is possible to attribute this to the data collection setting at a tertiary care facility, the more developed EMS system in the region due to the implementation of an ambulance service with funds from the Indian government in 2016, and public awareness campaigns that have promoted the new ambulance service [14]. As data collection was based in a large, tertiary teaching hospital, the number found in our study could be higher than what would be observed at a lower level hospital. Free ambulance transport could have also contributed to the higher proportion of ambulance utilization observed in our study, as cost of ambulance transport has been cited as a barrier to usage in other resource-limited settings [25,27,31].

Patients in the 56 to 95 age group had lower odds of using an ambulance compared to the 18 to 39 age group. This result conflicts with previous studies in resource-limited settings [27,30,31]. In our study, over half of the patients in the oldest age group were injured at home and injured by a fall. Both being injured at home and a fall were associated with lower odds of using an ambulance to travel to first health facility. If a patient was injured at home, they were more likely to travel to a health facility via a privately-owned vehicle than ambulance which

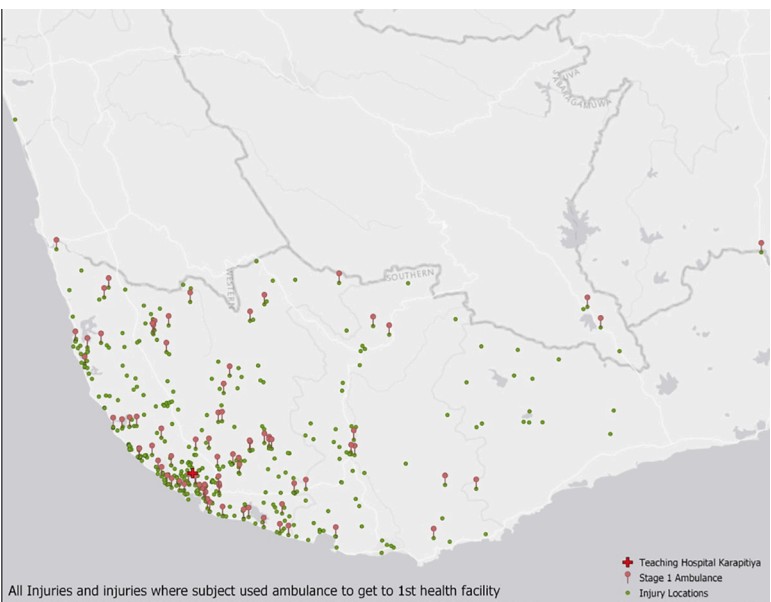

**Fig 1. Injury locations and ambulance transport.** Map depicting locations where injury event occurred (green dots, n = 400) and locations of injuries where patient took an ambulance to the first health facility (red pins, n = 83 of 400). The injury locations for 5 patients could not be mapped, thus n = 400.

could explain the discrepancy between this study and previous studies. Since the ambulance service is relatively new, there may be a lack of awareness among the older age group or a general lack of trust in the ambulance service [32]. Additionally, RTIs are the leading cause of death and injury among 15 to 29 year olds [3], and in our study patients in the 18 to 39 age group were more likely to be injured in a RTI which was associated with increased odds of ambulance transport. Both the higher rates of falls at home among the oldest age group and the increased frequency of RTI in the youngest age group could explain the discrepancy in findings between this study and other studies in resource limited settings.

Consumption of alcohol within six hours of the injury event was associated with significantly increased odds of ambulance transport in all models except the fully specified model. In the fully specified model, there was still increased odds of ambulance transport; however, the association was not significant. One explanation of this finding could be that of the patients who consumed alcohol within six hours of the injury, 46% were injured in a road traffic crash and 51% were injured in the street which were both associated with increased odds of using an ambulance to travel to the first health facility. Another potential explanation of this finding could be that patients who consumed alcohol within six hours of the injury event were, on average, more severely injured compared to those who did not consume alcohol which has been found in similar studies on acute injuries in resource-limited settings [33]. This could have resulted in more bystanders seeking medical help for the patient by calling an ambulance rather than transporting the patient in a private vehicle or tuk tuk. Other studies have shown that patients who arrive at the emergency department in an ambulance tend to have more severe injuries and poorer health outcomes [27,31,33].

## Strengths & limitations

Sri Lanka is currently in the process of developing an EMS system, but data on prehospital transportation and care is not systematically collected when a patient arrives at the hospital. Our study adds relevant information that could help inform next steps in the development of

**Table 4. Predictors of ambulance transport to first health facility: Logistic regression results.**

| | Odds Ratio (95% CI), P-Value | | | |
| --- | --- | --- | --- | --- |
| | **Bivariable** | | **Multivariable** | |
| **Age** | | | | |
| 18 to 39, REF | - | - | 0.70 (0.37, 1.32) | 0.273 |
| 40 to 55 | 0.68 (0.38, 1.20) | 0.183 | 0.55 (0.26,1.18) | 0.128 |
| 56 to 95 | 0.42 (0.22, 0.81) | 0.010* | | |
| **Gender** | | | | |
| Male, REF | - | - | 0.79 (0.43, 1.47) | 0.463 |
| Female | 0.61 (0.37, 1.02) | 0.058 | | |
| **Injury Mechanism** | | | | |
| RTI, REF | - | - | 1.62 (0.45, 5.89) | 0.464 |
| Fall | 0.36 (0.19, 0.66) | 0.001* | 0.58 (0.16, 2.08) | 0.402 |
| Stab/Cut | 0.28 (0.10, 0.75) | 0.011* | 0.85 (0.23, 3.10) | 0.808 |
| Other Blunt Force | 0.31 (0.13, 0.74) | 0.008* | 1.09 (0.21, 5.68) | 0.921 |
| Other | 0.42 (0.12, 1.50) | 0.181 | | |
| **Alcohol** | | | | |
| No, REF | - | - | 1.66 (0.85, 3.26) | 0.138 |
| Yes | 2.04 (1.16, 3.59) | 0.014* | | |
| **Location Type** | | | | |
| Street, REF | - | - | 0.29 (0.08, 0.98) | 0.047* |
| Home | 0.26 (0.14, 0.48) | <0.001* | 0.43 (0.12, 1.56) | 0.198 |
| Work | 0.44 (0.19, 1.01) | 0.052 | - | - |
| Market | - | - | 0.28 (0.06, 1.26) | 0.097 |
| Other | 0.44 (0.14, 1.36) | 0.154 | | |
| **Distance to THK (km)** | 1.00 (0.98, 1.01) | 0.725 | 1.00 (0.98, 1.01) | 0.575 |
| **Distance to Nearest Health Facility (km)** | 0.86 (0.69, 1.06) | 0.157 | 0.87 (0.69, 1.10) | 0.234 |
| **Open Wound** | 2.26 (1.19, 4.29) | 0.013* | 2.06 (1.00, 4.25) | 0.051 |
| No Open Wound, REF | - | - | | |
| **Abrasion** | 1.90 (1.05, 3.44) | 0.033* | 1.06 (0.52, 2.14) | 0.880 |
| No Abrasion, REF | - | - | | |
| **Chest/Abdomen** | 2.11 (1.03, 4.32) | 0.040* | 2.19 (0.94, 5.11) | 0.070 |
| No Chest/Ab Inj., REF | - | - | | |

*Denotes statistical significance where alpha<0.05.

EMS in Sri Lanka. There are several limitations of this study that should be noted. First, precise data on the amount of patients who refused to participate in the study was not collected. Rates of refusal to participate were generally low throughout data collection and it is estimated that no more than 20 people declined to participate. As such, we do not believe that refusal rates substantially altered our results and that our results are representative of those presenting to THK for injury care. Data collection was based out of a tertiary care hospital, thus introducing selection or sampling bias. Individuals who had more severe injuries and died before reaching the hospital or those who were receiving care in the intensive care unit at the time of data collection were not included in the study sample. Additionally, those patients whose injuries were treated at other health facilities and did not require referral to THK for specialist care or those who used an ambulance to travel to the hospital but were discharged the same day were excluded from the study. Thus, it is possible that the estimate of the proportion of ambulance transport to the first health facility obtained in this study is an overestimate of the true proportion. However, because THK is the largest tertiary care facility in the Southern province, it provided the most representative sample of prehospital transportation and care trends compared to if data collection was based out of a lower level health facility.

Generalizability may also be somewhat limited in that we could not collect data from the proxies of patients receiving care in the ICU which resulted in patients with the most severe

injuries being excluded from the study sample. The exclusion of the most severe injuries from this study could have led to an underestimation of ambulance usage, as severely injured patients are more likely to be transported to the hospital via ambulance [27]. While this exclusion may have affected the proportion of ambulance usage we obtained, it is unlikely to substantially change our logistic regression model results. A prior population-based injury study in the Galle district found that RTIs and falls accounted for most injury related deaths [4]; and given our results mirror this, we believe that even without the most severely injured ICU patients our population represents the patients presenting to THK for injury care.

## Implications for future research

A free ambulance service was launched in 2016 and our study showed that 20.5% of patients treated at THK for an injury used an ambulance to travel to the first health facility. Future research on the prehospital transportation and care of those with externally-caused, acute injuries should include those with more severe injuries to increase the generalizability of the results and to better understand the use of ambulances. Future studies should also examine in more detail the characteristics of those who use an ambulance, the type of injuries that affect patients who use an ambulance, and the type of care provided by emergency personnel and whether that care is effective in improving patient outcomes. Understanding the characteristics of those who use an ambulance as well as the relationship between ambulance use and patient outcomes would aid in identifying strengths in the current EMS system as well as areas for improvement. The goal of this line of research is to improve access to effective and timely EMS services for the patients of Sri Lanka so that the burden of injuries may be reduced.

## Conclusion

In this study conducted in a lower acuity patient population, the results show that 20.5% of patients used an ambulance to travel to the first health facility, while over half of patients traveled in a tuk tuk. Additionally, older age, injury mechanism and being injured at home were significantly associated with lower odds of ambulance transport, while distance from injury location to the nearest health facility or THK was not associated with ambulance transport to the first health facility. Future research should include patients with more severe injuries, collect data on care provided during transport and examine the association between ambulance transport and patient outcomes. This research may help in developing strategies to reach those populations who are not using ambulances but should be transported via ambulance given the critical nature of their injuries.

## Supporting information

**S1 File. Anlaysis data.** This is the de-identified dataset used in all analyses contained within this manuscript.
(CSV)

**S2 File. Data collection questionnaire (English).** This is the English version of the survey that was used to collect the data for this analysis.
(DOCX)

## Author Contributions

**Conceptualization:** Lindy M. Reynolds, Vijitha De Silva, Shayna Clancy, Catherine A. Staton, Truls Østbye.

**Data curation:** Lindy M. Reynolds, Vijitha De Silva.

**Formal analysis:** Lindy M. Reynolds, Anjni Joiner, Catherine A. Staton, Truls Østbye.

**Methodology:** Vijitha De Silva, Anjni Joiner, Catherine A. Staton, Truls Østbye.

**Project administration:** Lindy M. Reynolds, Vijitha De Silva, Truls Østbye.

**Resources:** Shayna Clancy.

**Software:** Lindy M. Reynolds.

**Supervision:** Lindy M. Reynolds, Vijitha De Silva, Shayna Clancy, Anjni Joiner, Catherine A. Staton, Truls Østbye.

**Writing – original draft:** Lindy M. Reynolds, Anjni Joiner.

**Writing – review & editing:** Lindy M. Reynolds, Vijitha De Silva, Shayna Clancy, Anjni Joiner, Catherine A. Staton, Truls Østbye.

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
