## [Decision Letter · Decision Letter 0]

19 Apr 2021

PONE-D-21-02599

Predictors of ambulance transport to first health facility among injured patients in Southern Sri Lanka

PLOS ONE

Dear Dr. Reynolds,

Thank you for submitting your manuscript to PLOS ONE. After careful consideration, we feel that it has merit but does not fully meet PLOS ONE’s publication criteria as it currently stands. Therefore, we invite you to submit a revised version of the manuscript that addresses the points raised during the review process.

I apologize for the delay and any inconvenience it may have caused. It was indeed a very difficult task to find reviewers. Finally, I got two reviews. Reviewer 2 is a very proficient expert of the "International Committe of the Red Cross".

We look forward to receiving your revised manuscript.

Kind regards,

Hans-Peter Simmen, M.D., Professor of Surgery

Academic Editor

PLOS ONE

Journal Requirements:

3. In your Methods section, please provide a justification for the sample size used in your study, including any relevant power calculations (if applicable).

4. We note that Figure 1 in your submission contain map images which may be copyrighted. All PLOS content is published under the Creative Commons Attribution License (CC BY 4.0), which means that the manuscript, images, and Supporting Information files will be freely available online, and any third party is permitted to access, download, copy, distribute, and use these materials in any way, even commercially, with proper attribution. For these reasons, we cannot publish previously copyrighted maps or satellite images created using proprietary data, such as Google software (Google Maps, Street View, and Earth). For more information, see our copyright guidelines: http://journals.plos.org/plosone/s/licenses-and-copyright.

4.1.    You may seek permission from the original copyright holder of Figure 1 to publish the content specifically under the CC BY 4.0 license. 

4.2.    If you are unable to obtain permission from the original copyright holder to publish these figures under the CC BY 4.0 license or if the copyright holder’s requirements are incompatible with the CC BY 4.0 license, please either i) remove the figure or ii) supply a replacement figure that complies with the CC BY 4.0 license. Please check copyright information on all replacement figures and update the figure caption with source information. If applicable, please specify in the figure caption text when a figure is similar but not identical to the original image and is therefore for illustrative purposes only.

Reviewers' comments:

Reviewer's Responses to Questions

**Comments to the Author**

1. Is the manuscript technically sound, and do the data support the conclusions?

Reviewer #1: Partly

Reviewer #2: Partly

2. Has the statistical analysis been performed appropriately and rigorously? 

Reviewer #1: N/A

Reviewer #2: Yes

3. Have the authors made all data underlying the findings in their manuscript fully available?

Reviewer #1: Yes

Reviewer #2: Yes

4. Is the manuscript presented in an intelligible fashion and written in standard English?

Reviewer #1: Yes

Reviewer #2: Yes

5. Review Comments to the Author

Reviewer #1: The article is written in appropriate English but the scientific conclusions are not sufficient. An observational study of prehospital care of only 405 patients in an tertiary care hospital is difficultly representative in numbers and quality.

Reviewer #2: Comments to the Author:

Things to be considered before publishing:

• The paper does not provide information about nbrs of patients who did not want to participate

• The missing (n=405 vs n=400) 5 patients from whom a location could not be determined should just be mentioned for completeness

6. PLOS authors have the option to publish the peer review history of their article (what does this mean?). If published, this will include your full peer review and any attached files.

Reviewer #1: No

Reviewer #2: **Yes: **Dr. Thomas Wilp, Ph.D., MPH, MBA, MDM, M.A.

---

## [Author Response · Author response to Decision Letter 0]

1 Jun 2021

Journal Requirements:

We have carefully reviewed the journal’s style requirements, including those for file naming, and have updated the following files to conform to the requirements:

Manuscript

Revised Manuscript with Track Changes

Fig1

S1 File

S2 File

The survey is not copyrighted, and we have now included the survey in English as a supplementary file (S2 File). 

3. In your Methods section, please provide a justification for the sample size used in your study, including any relevant power calculations (if applicable).

Response: This study was designed to gather baseline information on prehospital transportation and care trends of injured patients at the main referral hospital in southern Sri Lanka. Due to logistical concerns, we were limited in our ability to justify a sample size. The resulting sample size is what we were able collect over a two-month period given the injured patient volume at this hospital, after accounting for the low numbers of patients who refused to participate in our study. 

4. We note that Figure 1 in your submission contain map images which may be copyrighted. 

Response: Figure 1 was created by the authors using ArcGIS Pro, a desktop geographic information system application that supports viewing, editing, and analysis of geospatial data. This figure is not copyrighted and was instead generated by the authors.

5. Please include captions for your Supporting Information files at the end of your manuscript, and update any in-text citations to match accordingly. 

Response: We have now added captions for the supporting information files at the end of the manuscript. These captions begin on line 420. 

Review Comments to the Author

Reviewer #1: 

The article is written in appropriate English but the scientific conclusions are not sufficient. An observational study of prehospital care of only 405 patients in a tertiary care hospital is difficultly representative in numbers and quality.

Response: Thank you for this comment. This study was designed to gather baseline information on prehospital transportation and care trends of injured patients that is not regularly collected elsewhere. The hospital where our study was conducted is the only hospital in the region where all of the specialties are available, including orthopedic surgery. Thus, we believe that even with a lower sample size our study sample is representative of these trends because all injured patients requiring any specialty care would likely end up at the hospital where data collection was based and would have been approached about participation in the current study. 

Reviewer #2: 

The paper does not provide information about numbers of patients who did not want to participate.

Response: Thank you for noting this important piece of information that is omitted from this manuscript. While we did not collect an exact number of patients that refused to participate, the research assistants noted very small numbers of patients refusing to participate throughout the 2 months of active data collection. It is estimated that between 15 and 20 patients over the course of data collection refused to participate. We do not believe this number would have substantially altered our results. We have added 3 sentences in the strength & limitations section regarding this (lines 284-289).

The missing (n=405 vs n=400) 5 patients from whom a location could not be determined should just be mentioned for completeness. 

Response: Thank you for this comment. The figure caption has been updated to include a sentence about the missing locations in lines 201-202. Additionally, a sentence has been added to the body of the results (lines 191-193) noting the missing 5 locations and sample size for logistic regression models where distance was included as a predictor.

The results of the study that 20.5% of patients used ambulance transportation to get to the first health facility in undoubtable a valuable source for information for prehospital emergency care providers in LMIC but does not represent scientifically unknown or surprising new knowledge. 

Response: Thank you for this comment. We agree that the percent of patients who used an ambulance to get to the first health facility provide a valuable source of information for prehospital care providers and those professionals involved in health care and emergency medical system planning. Because data on prehospital care and transportation are not regularly collected in the Sri Lankan context, we believe our study adds valuable and previously unknown information for local officials. Additionally, availability and usage of ambulances in LMIC is significantly variable, thus these results add to the overall body of literature on prehospital care in LMICs.

The factors that the researchers associated with ambulance use are of value for comparisons as these are the most often used indicators in the western world. Unfortunately do these westernized indicators (as well as most western researchers) often neglect the context the subject of research Emergency Medical Service (EMS) is imbedded in, and the obstacles that LMIC often present when it comes prehospital service implementation.

Some examples of obstacles:

• Is the ambulance service known by the affected patients (families/ bystanders/ population)? 

• Is the EMS accessible and known (does a unified emergency nbr exist; is it really free or are hidden charges existing….)

• Are there laws and regulations existing (right of way for ambulances (traffic jams); lights & sirens use) and therefore is the really the fastest means transport (or are tuk-tuks much faster)?

• Is there any recognition (scope of practise/license) of care takers existing and to they really care (stabilize) for patients during transportation? (or may it be forbidden by law?)

• In case of RTI do emergency services like fire & police activate/ call EMS?

Response: Thank you for raising these important questions. While our survey did not collect data about awareness about the ambulance service, there were documented public awareness campaigns about the new service and we believe that people widely knew about the service when our study was conducted. A unified emergency number exists (#: 1990) and it is truly free with no hidden charges. There are laws and regulations regarding right of way, sirens and lights with ambulances. It is generally considered to be the fastest mode of transportation; however, there is a wait time for the ambulance to arrive to the scene of the injury where tuk-tuks are often more readily available. People may opt to take a tuk-tuk because they could be on their way to the hospital as opposed to waiting for the ambulance to arrive. There is a recognition of the scope of practice of the personnel who work the ambulance. The ambulances are often staffed by a minimum of 1 person who received intensive training for 3 months and another person who is trained in basic life support and first aid. The paramedics are able to provide medical care while in route to the hospital. In the case of RTI, police are usually involved following the accident but fire is not typically involved unless it was a major accident. The second paragraph of the introduction has been updated with this information (lines 51-81).

Some of these highly important sources of information the researchers looked into, others where limited due to the fact that the research was a hospital-based cross-sectional study. The other limitations of the study were explained well and the most limiting ones (patients younger than 18years and severely injured patients) could have probably been extrapolated or data could have been gathered later or from care takers (parents). Patients who rejected participating in the study are unfortunately not mentioned.

Response: We agree that our study was limited by the fact that we could not collect data from patients younger than 18 years and that we could not collect data from patients receiving care in the intensive care unit. We have added a few sentences, as noted above, regarding the patients who refused to participate in our study (lines 284-289). 

The conclusion that among lower acuity injury patients in southern Sri Lanka, 20.5% travelled in an ambulance to the first health facility, while others used other means of transport (over 50% used tuk tuks) is devastation for a young and prospering ambulance service but can serve well for further research needed to find out about the reasons behind. The predictors used by the researcher provide good information but may not represent the main barriers of EMS use. 

Response: We agree that the low percentage of those that chose to take an ambulance to the first health facility is concerning for local health system and emergency officials. Investigating the reasons for why people make certain prehospital transportation and care choices is a very important area of research especially in places where the ambulance service is new. Unfortunately, this was beyond the scope of the current project, but we believe it would be worth pursuing in future studies to improve understanding of the barriers to EMS use. We do not believe a lack of awareness or cost contributed to the lower percentage we obtained in this study, but it may be that tuk-tuks are more readily available means of transportation and people can begin traveling to the hospital sooner as opposed to waiting on the ambulance to arrive. We believe our study adds valuable baseline information that can inform future studies on the ambulance service and EMS system with the ultimate goal of improving patient care and outcomes.

---

## [Editor Report · Decision Letter 1]

7 Jun 2021

Predictors of ambulance transport to first health facility among injured patients in Southern Sri Lanka

PONE-D-21-02599R1

Dear Dr. Reynolds,

We’re pleased to inform you that your manuscript has been judged scientifically suitable for publication and will be formally accepted for publication once it meets all outstanding technical requirements.

Kind regards,

Hans-Peter Simmen, M.D., Professor of Surgery

Academic Editor

PLOS ONE
---

## [Editor Report · Acceptance letter]

18 Jun 2021

PONE-D-21-02599R1 

Predictors of ambulance transport to first health facility among injured patients in southern Sri Lanka 

Dear Dr. Reynolds:

I'm pleased to inform you that your manuscript has been deemed suitable for publication in PLOS ONE. Congratulations! Your manuscript is now with our production department. 

Kind regards, 

on behalf of

Dr. Hans-Peter Simmen 

Academic Editor

PLOS ONE